# Assessing the Benefit of Dietary Choline Supplementation Throughout Adulthood in the Ts65Dn Mouse Model of Down Syndrome

**DOI:** 10.3390/nu16234167

**Published:** 2024-11-30

**Authors:** Savannah Tallino, Rachel Etebari, Ian McDonough, Hector Leon, Isabella Sepulveda, Wendy Winslow, Samantha K. Bartholomew, Sylvia E. Perez, Elliott J. Mufson, Ramon Velazquez

**Affiliations:** 1Banner Neurodegenerative Disease Research Center, Biodesign Institute, Arizona State University, Tempe, AZ 85281, USA; stallino@asu.edu (S.T.); retebari@asu.edu (R.E.); itmcdono@asu.edu (I.M.); hjleon@asu.edu (H.L.); issepulv@asu.edu (I.S.); wendy.winslow@asu.edu (W.W.); skbartho@asu.edu (S.K.B.); 2School of Life Sciences, Arizona State University, Tempe, AZ 85287, USA; 3Barrow Neurological Institute, Phoenix, AZ 85013, USA; sylvia.perez@commonspirit.org (S.E.P.); elliott.mufson@barrowneuro.org (E.J.M.); 4Arizona Alzheimer’s Consortium, Phoenix, AZ 85014, USA

**Keywords:** down syndrome, choline, diet supplementation, inflammation, cognition

## Abstract

Background/Objectives: Down syndrome (DS) is the most common cause of early-onset Alzheimer’s disease (AD). Dietary choline has been proposed as a modifiable factor to improve the cognitive and pathological outcomes of AD and DS, especially as many do not reach adequate daily intake levels of choline. While lower circulating choline levels correlate with worse pathological measures in AD patients, choline status and intake in DS is widely understudied. Perinatal choline supplementation (Ch+) in the Ts65Dn mouse model of DS protects offspring against AD-relevant pathology and improves cognition. Further, dietary Ch+ in adult AD models also ameliorates pathology and improves cognition. However, dietary Ch+ in adult Ts65Dn mice has not yet been explored; thus, this study aimed to supply Ch+ throughout adulthood to determine the effects on cognition and DS co-morbidities. Methods: We fed trisomic Ts65Dn mice and disomic littermate controls either a choline normal (ChN; 1.1 g/kg) or a Ch+ (5 g/kg) diet from 4.5 to 14 months of age. Results: We found that Ch+ in adulthood failed to improve genotype-specific deficits in spatial learning. However, in both genotypes of female mice, Ch+ significantly improved cognitive flexibility in a reverse place preference task in the IntelliCage behavioral phenotyping system. Further, Ch+ significantly reduced weight gain and peripheral inflammation in female mice of both genotypes, and significantly improved glucose metabolism in male mice of both genotypes. Conclusions: Our findings suggest that adulthood choline supplementation benefits behavioral and biological factors important for general well-being in DS and related to AD risk.

## 1. Introduction

Down syndrome (DS) occurs in 1/700 live births and is the most common cause of early-onset Alzheimer’s disease (AD) [1,2]. The dramatic increase in life expectancy for individuals with DS—from a median of 10 years in the 1970s to 65 years currently—revealed that nearly all will develop AD proteinopathies, including amyloid beta (Aβ) and neurofibrillary tangles of hyperphosphorylated tau, by 40–50 years of age [3,4]. As the general population ages, it is projected that by 2060, nearly 14 million Americans over the age of 65 will have AD [5]. However, the pathobiology of AD likely begins decades prior to noticeable age-related cognitive decline. For example, MRI-based studies have shown that, prior to Aβ deposition, basal forebrain regions rich in cholinergic neurons (BFCNs) such as the nucleus basalis of Meynert (NBM) begin to atrophy by ~20 years of age in DS and during mild cognitive impairment in prodromal AD [6,7]. BFCNs from the NBM and the medial septum (MS) provide the major source of cholinergic innervation to the frontal cortex and hippocampus,, modulating learning, memory, and executive functions such as attention [8]. Thus, there is an urgent need to develop novel treatment options that may be implemented earlier in life to protect against BFCN dysfunction during the development of AD in individuals with DS as well as within the general aging population. Risk factors that may underlie cellular degeneration in AD likely include hypertension, type II diabetes (T2D), obesity, and dyslipidemia, with peripheral inflammation playing a central role in linking these disorders with AD [9,10,11,12]. For example, higher mid-life systemic inflammation has been linked to reduced brain volumes in key regions vulnerable to AD-related atrophy [13]. This is particularly concerning given the high rates of obesity, T2D, and increased peripheral inflammation in DS [14,15]. Dietary modifications may be a key strategy to attempt to target these pathologies prophylactically and, thus, reduce the risk of AD. To this end, multivitamin supplementation has been utilized in clinical trials to demonstrate cognitive improvements in older adults [16], and a recent review of findings in older adults both with and without AD has suggested a specific list of vitamins, minerals, and other supplements that may help support cognitive functions [17].

Dietary choline intake has been investigated as a modifiable factor relevant to both AD and DS, particularly given its role in the synthesis of acetylcholine for BFCN neurotransmission and given the multiple lines of evidence supporting its neuroprotective role (reviewed previously [18]). Endogenous choline can be synthesized in the liver, with its metabolism central to multiple cognition-relevant pathways—both directly and as a methyl-group donor via betaine—including acetylcholine synthesis, membrane phospholipid synthesis, epigenetic regulation via DNA methylation, and methionine and folate metabolism [18,19]. However, endogenous choline synthesis is insufficient for the body’s needs such that a recommended adequate intake (AI) was set in 1998 to prevent hepatic damage [18,19]. The AI for choline in adults is 550 mg/day for men and 425 mg/day for women (increasing to 450 and 550 mg/day if pregnant or lactating, respectively); however, most people worldwide do not reach the AI for choline, including around 90% of the US population, and low choline intake often correlates with racial and socioeconomic disparities [20,21,22]. This situation is alarming in the context of the burgeoning rise in AD, given that we have shown previously in AD mouse models that inadequate dietary choline exacerbated AD pathology and also that low circulating choline levels in humans correlated with high pathological AD burden [23,24]. Methylation markers in the blood suggest some dysregulation in choline-related metabolic pathways in DS [25], which suggests the need to determine whether the DS population meets dietary choline levels.

Considering the accelerated timeline of AD development in DS, and due to the heightened choline requirements for pregnant and lactating mothers [20], multiple studies of choline supplementation have been performed during the perinatal period in Ts65Dn mice, the most commonly used rodent model of DS (Jackson Laboratory, Bar Harbor, ME, USA, Strain #005252; [26]). Specifically, past work has shown that maternal choline supplementation (MCS) benefitted Ts65Dn offspring by preventing genotype-driven loss of BFCNs [27,28], by improving septohippocampal-dependent behaviors [27,29], and by improving other measures of BFCN-mediated attentional performance [30,31,32]. Recent work has also shown that MCS in trisomic Ts65Dn mice normalized changes in mRNA transcripts within the BFCNs of the MS and vertical diagonal band (VDB) [33] and in hippocampal cornu ammonis 1 (CA1) neurons [34,35], improved the dysregulation of metabolism-related proteins in the frontal cortex [36], and increased hippocampal neurogenesis [29]. Although the beneficial effects of MCS persisted despite the discontinuation of choline supplementation after weaning, there was a diminished effect with aging [32]. MCS has also been studied in AD mouse models, where the offspring of supplemented dams displayed improved spatial memory and less Aβ plaque pathology [37,38].

Although the impact of initiating a choline-supplemented diet in adulthood has been less studied in AD mouse models, adulthood dietary choline supplementation ameliorated AD pathology and improved spatial learning and memory in males [39] and females [40] of the APP/PS1 mouse model of AD. However, whether a postnatal choline-supplemented diet can affect BFCNs or septohippocampal-dependent learning and memory deficits in trisomic Ts65Dn mice of both sexes remains an under-investigated area. In this regard, others have suggested that not only is further research needed in this area, but studies should include both male and female animals [41]. Here, we assessed the effect of choline supplementation in male and female Ts65Dn mice, starting at an average age of 4.7 months (mo)—which is before BFCN morphological alterations commence in this strain [42]—and continuing until animals were euthanized at an average age of 14.3 mo. We hypothesized that (1) if adulthood supplementation did not show the same benefit to BFCNs and BFCN-mediated behavior as perinatal supplementation, this would support the need for early supplementation in DS; and (2) adulthood supplementation would improve other features relevant to the risk of AD. Therefore, we assessed a variety of metabolic and behavioral outcomes including those benefited by MCS, as well as other peripheral pathologies relevant in DS and AD.

## 2. Materials and Methods

### 2.1. Animals

All protocols were approved by the Institutional Animal Care and Use Committee of Arizona State University (protocol number 22-1933R) and conform to the National Institutes of Health Guide for the Care and Use of Laboratory Animals. For this study, we used Ts65Dn mice (B6EiC3Sn.BLiA-Ts(1716)65Dn/DnJ; Jackson Laboratory, Bar Harbor, ME, USA, Strain #005252), which have been described previously [43,44]. Trisomic (3n) Ts65Dn mice carry a translocation chromosome that combines murine chromosome 16 and part of 17, with approximately 65% of the *HSA21* orthologous genes [45,46]; their disomic (2n) counterparts are used as controls. The strain is maintained by breeding 3n dams to males of the B6EiC3Sn.BLiAF1/J background (Jackson Laboratory Strain #003647); this strain lacks the recessive *Pde6b^rd1^* mutation of earlier background strains, which avoids the complication of retinal degeneration and blindness in a subset of pups [44]. Founder 3n dams were purchased in 2019 from Jackson Laboratory and the colony was supplemented with additional breeder dams from Jackson Laboratory at least once per year to reduce founder effects and to avoid phenotypic drift, as suggested recently [47]. 3n mice and 2n littermates were same-sex group housed (4–5 mice per cage) after weaning at 21 days of age; all cages contained enrichment in the form of mouse igloos, bedding nestlets, and small nylon bones. For genotyping, tail snips were digested in sodium-dodecyl-sulfate Tris ethylene-diamine-tetra-acetic acid (EDTA) buffer with proteinase K, followed by DNA precipitation in isopropanol; genotypes were confirmed using the following polymerase chain reaction primers:Ts65Dn Forward: GTGGCAAGAGACTCAAATTCAACTs65Dn Reverse: TGGCTTATTATTATCAGGGCATTTInternal Control Forward: CTAGGCCACAGAATTGAAAGATCTInternal Control Reverse: GTAGGTGGAAATTCTAGCATCATCC

At 4–5 mo of age (average age 4.7 mo)—prior to when basal forebrain pathology in 3n mice is documented to emerge [42]—mice were switched from standard laboratory chow to modified AIN-76A (CA.170481) from Envigo Teklad Diets (Madison, WI, USA; Appendix A). Mice received either 1.1 g/kg choline chloride (Choline Normal, Envigo Teklad Diets, Madison, WI, USA, TD.140777, ChN;) or 5 g/kg (Choline Supplemented, Envigo Teklad Diets, Madison, WI, USA, Ch+; TD.140778, Ch+). Mice were fed these diets for the duration of the study until euthanasia (14–15 mo, average 14.3 mo), for a total of 9.5 mo. Animals were weighed prior to the start of ChN and Ch+ diets and every 2 weeks thereafter for the duration of the study. At 11.5 mo of age, food intake was weighed by cage for 7 consecutive days to assess intake levels across the groups. Behavior testing began at 12.7 mo. During IntelliCage testing, 4 males had to be euthanized early due to severe fight wounds, per institutional protocol, and 9 males were treated by veterinary staff for mild-to-moderate fight wounds. As such, all males were returned to regular cages to allow recovery and prevent further fighting until the end of the study. Attrition with age was as expected in this strain, and all remaining mice were euthanized at the end of study.

### 2.2. Behavior

#### 2.2.1. Rotarod

Rotarod testing was used to assess motor function and coordination as previously described [23]. Briefly, mice were placed on a rod that accelerates (AccuRotor Rota-Rod, Omnitech Electronics, Columbus, OH, USA) and latency to fall was measured as the output representing motor ability. During the first 2 consecutive days (training days), mice were tested for 6 trials with the rod accelerating at 0.75 rpm per second (s) for 20 s until it reached 15 rpm, where it remained constant for a total of 90 s. On the 3rd day (probe day), the rod continuously accelerated 1 rpm/s for 90 s.

#### 2.2.2. Radial Arm Water Maze (RAWM)

RAWM testing was used to assess septohippocampal-dependent spatial learning and memory [48]. The 8-armed maze (described previously [49]) contains a wide platform in one arm, referred to hereafter as the goal arm. The container was filled with water such that the platform was submerged by exactly 1 cm, and water was kept between 23–24 °C as Ts65Dn mice are particularly susceptible to hypothermia at lower temperatures but require adequate motivation to find the platform [50]. Water was made opaque with addition of white non-toxic craft paint to hide the platform location, forcing mice to rely on external maze cues placed on the walls surrounding the apparatus; all other objects in the room, including experimenter position, were maintained exactly throughout all trials. On the 1st day only, mice were placed on the platform for 10 s to orient them to the location of the goal arm prior to starting the testing procedure. Following that orientation, each mouse was tested in 4 consecutive trials per day for 6 days. Trials began in pseudo-randomized entry arms, where mice were placed in the arm facing the maze center, and allowed to swim freely for 60 s. Subsequent arm entries were recorded manually and determined as when the base of the animal’s tail entered the arm threshold. Trials were also recorded in case later review of entries was necessary. If the animal reached the platform in the goal arm and climbed upon it, the latency to find the platform was recorded and the animal was allowed to rest there for 10 s to reinforce the spatial cues surrounding the location; otherwise, if an animal failed to find the platform by 60 s, they were gently guided to the platform and allowed to rest for 10 s. After each trial, the animal was placed in a warming cage for 25 s prior to initiation of the subsequent trial. Following the 4th trial, the animal was gently dried and placed in a separate warming cage until completely dried, then returned to its home cage until the next day’s trials. Mice unable to swim (defined as the inability to keep their head above water) were excluded from all subsequent trials. Dependent measures on the RAWM task included latency to platform, arm entry errors (entries into an arm that was not the goal arm, a measure of reference memory), arm repeat entry errors (re-entries into a previously-visited arm, a measure of working memory), the number of trials in which the platform was found, and chaining events (3 consecutive entries into adjacent arms in a clockwise or counter-clockwise pattern). To control for mice with a low number of arm entries that did not find the platform (as a low number of arm entries may appear as good performance, but not if the animal did not find the platform), we calculated the ratio of correct arm entries to total arm entries, with a scale of 0–1 indicating performance (1 = a perfect trial, 0 = a trial where the animal never entered the goal arm).

#### 2.2.3. IntelliCage

We used the IntelliCage automated behavioral phenotyping system to assess exploration, reference memory, and cognitive flexibility [51]. Prior to IntelliCage testing, animals were implanted with a radiofrequency identification transponder chip (RFID; Standard Microchip T-VA, DataMars, Switzerland & Troven, USA) to allow for identification in the IntelliCage system. RFIDs were implanted in the subcutaneous dorso-cervical region under isoflurane inhalation anesthesia as described previously [51,52] and were given a minimum of 7 days to recover prior to testing. A maximum of 16 animals were co-housed per cage, over the course of 18 days, for a battery of behavioral tasks. IntelliCage testing (described previously [51,52]) uses animals’ motivation to consume water to drive learning of operant testing where access to water can be made dependent on learned behaviors. The operant corners include an RFID sensor, so that each animal’s unique RFID is logged upon entering a corner, and a water access port including a nosepoke sensor, doors, LED lights, and a lickometer. Performance readouts include corner entries, nosepokes, and licks. Animals who fail to drink within a 24 h period were removed, per institutional protocol.

The IntelliCage tasks were as follows:General Adaptation phase (2 days): By default, doors in each access port were consistently open, and animals were allowed to drink in all 4 corners of the cage 24 h per day. Total corner visits and total licks were measured.Door Adaptation phase (2 days): By default, doors in each access port were closed. When entering a corner, RFID presence signaled the doors to open, allowing access to water. Water was accessible 24 h per day. Total corner visits and total licks were measured.Nosepoke Adaptation phase (2 days): When entering a corner, RFID presence followed by a nosepoke signaled the door to open, allowing access to water. Water was accessible 24 h per day. Total corner visits and total licks were measured, as were corner visits with ≥1 nosepoke, and corner visits with ≥1 lick.Place Preference (6 days): Animals were assigned a corner, and entry into that corner with a nosepoke signaled the access port doors to open only if the animal was in their assigned corner; nosepokes in any other corner did not allow water access. Water was accessible in the assigned corner 24 h per day. In addition to total licks and total corner visits, we measured visits to the assigned corner, visits with ≥1 nosepoke, and visits with ≥1 lick. We also calculated percent assigned correct as the number of visits to assigned corner/total corner visits, and percent correct visits as the number of assigned visits with >1 lick/total corner visits.Reverse Place Preference (6 days): The water access ports in the assigned corner from the previous task were no longer accessible; instead, the corner opposite the previously assigned corner became the newly assigned corner, and we measured the same outputs as in the previous task with the addition of the percentage of visits to the previously assigned corner. Water was accessible in the assigned corner 24 h per day.

Data were extracted via the TSE IntelliCagePlus Analyzer software followed by conversion into a single SQLite3 database file using Python (Version 3.12). Additional Python scripts queried the database using SQL to group data into 24 h periods, extract relevant dependent variables as defined above, and export into Microsoft Excel.

### 2.3. Glucose Tolerance, Euthanasia, and Peripheral Measures

#### 2.3.1. Glucose Tolerance Testing

Following behavior testing, glucose tolerance testing was performed 1 week prior to euthanasia using the TRUEtrack glucose meter and testing strips (Trividia Health, Fort Lauderdale, FL, USA, A3H0180) as described previously [23,52]. After overnight fasting (16 h), mice were weighed, baseline fasting blood glucose was sampled from the tail, and then, all animals were injected intraperitoneally with 2 mg/kg bodyweight of glucose. Blood glucose was monitored from the tail at 15, 30, 45, 60, 90, and 120 min after injection.

#### 2.3.2. Euthanasia and Tissue Collection

At 14.3 mo, after a total of 9.5 mo on the experimental diets, animals were weighed and anesthetized with ketamine and xylazine (120 mg/kg and 6 mg/kg body weight, respectively) before transcardial perfusion with phosphate-buffered saline (PBS). Whole brains and a portion of the left liver lobe were collected and immediately fixed in cold 4% paraformaldehyde in PBS. After 48 h of fixation at 4 °C, tissue was transferred to 0.02% sodium azide in PBS for long term storage at 4 °C.

#### 2.3.3. Blood Plasma Collection and Analyses

Blood plasma was collected at baseline (4.7 mo) and at euthanasia (14.3 mo), following a period of fasting for approximately 16 h. Blood was collected via submandibular vein puncture into sterile tubes containing EDTA (BD, Franklin lakes, NJ, USA, K2EDTA #365974) to prevent clotting, incubated on ice for 60 min, then centrifuged for 30 min at 4 °C at 455× *g*. Supernatant (plasma) was collected and stored long term at −80 °C before being used for analyses of circulating total choline levels (Abcam, Waltham, MA, USA,ab219944) and cytokine levels via 23-plex kit (Bio-Rad, Hercules, CA, USA,M60009RDPD). Plasma choline was measured using 20 μL of plasma per animal, in duplicate. The multiplexed cytokine measurements were performed using 15 μL of plasma per animal, in duplicate as technical replicates, and analyses were performed using the Bio-plex^®^-200 and Bioplex Manager software (Version 6.2) [53]. Technical replicates were averaged for each animal. Cytokine levels that were not detected were not included in the analysis, and levels between the blank and lowest standard were extrapolated using the standard curve equation for that particular cytokine.

#### 2.3.4. Liver Histology

To assess the effects of choline supplementation on hepatic pathology, livers were sectioned using a vibratome (Leica, Wetzlar, GER, VT1000 s) at a thickness of 50 μm and then stained with hematoxylin and eosin using a commercially available kit (Abcam; ab245880), with images obtained using a light microscope at 20× (Zeiss Axio, Oberkochen, GER, Imager M2). Scoring criteria for steatosis (fat accumulation) was as previously described [23,54], and sections were scored by 2 researchers blinded to subject groups during analysis and whose results were averaged for statistical analysis.

### 2.4. Immunohistochemistry and Unbiased Stereology of Basal Forebrain Cholinergic Neurons

Whole brains were embedded in 3% agarose in 1x PBS and sectioned coronally using a vibratome (Leica VT1000 s) at a thickness of 50 μm. To identify BFCNs, collaborators at the Barrow Neurological Institute performed immunohistochemistry on sections spanning the medial septum (MS) and vertical diagonal band (VDB), using the choline acetyltransferase (ChAT) antibody (Sigma Aldrich, St. Louis, MO, USA; AB144P) at a 1:800 dilution as previously described [55]. To quantify BFCNs, unbiased stereology was performed using MBF Bioscience Stereo Investigator software (version 2023). The optical fractionator method allows for a 3D space to be counted using a 2D grid and provides a start that is both unbiased and random. Preliminary work ensured that m = 1 Gundersen scores remained <0.1 and resulted in the following optimized stereology parameters: 20% of the region area sampled, dissector grid dimensions of 150 × 50 µm, and a 16 µm average dissector height. The MS and VDB regions were anatomically combined into 1 region for analysis as it is difficult to consistently demarcate these subregions across all tissue sections and subjects and because no significant differences have been observed when these subregions are combined [31]. Analysis was performed on n = 3–4 animals per diet and genotype, for each sex. A total of 6 sections per animal were sampled moving from rostral to caudal, with the 1st section of each sampling series containing an approximate Bregma coordinate of 0.98 mm, at an interval of n = 2 (every other section), until approximately 0.38 mm rostral to Bregma, resulting in a total z-plane depth of 550 µm. For consistency, a single investigator performed unbiased stereology and remained blinded to the subject groups for the entirety of the analysis.

### 2.5. Statistical Analysis

Multivariate analysis of variance (MANOVA) was carried out in IBM SPSS (version 28.0) to assess main effects of genotype, diet, sex, and—for repeated measures in behavioral testing—time, as MANOVA is robust against violations of sphericity. Significant interactions (*p* < 0.05) were probed by follow-up testing with Bonferroni corrections for multiple comparisons (alpha level of 0.05/number of comparisons). GraphPad Prism (version 10.2) was also used for Pearson’s correlation analyses, outlier testing using the ROUT method, and area under the curve calculations. Bioplex Manager software (Version 6.2) was used for outlier testing of multiplexed cytokine measurements. Note that males were removed from IntelliCage testing due to aggression, fight wounds, and subsequent veterinary treatment. Because the handling of males and females diverged significantly at this point, all statistical comparisons of data following this divergence were carried out to compare the effects of genotype and diet in males and females separately, with the exception of a single planned evaluation of BFCN cell density collapsed by sex for comparison to previous Ts65Dn literature.

## 3. Results

### 3.1. Choline Supplementation Decreases Weight Gain in Females Regardless of Food Intake and Genotype

Sixty-seven animals (n = 7–10 per sex, diet, and genotype) began experimental diets (Appendix A) of either 1.1 g/kg choline (Choline Normal; ChN) or 5 g/kg choline (Choline supplemented; Ch+) at an average age of 4.7 mo (Figure 1A,B). Trisomic (3n) animals were compared to their disomic (2n) littermate controls. At 11.5 mo of age, when animals had been on the diets for 7 mo, we assessed average food intake per cage, adjusted by the weight of mice per cage (n = 4–5 cages per sex and diet; (Figure 1C)) and observed that males consumed more food than females (F_(1, 13)_ = 9.06; *p* = 0.01), but Ch+ did not affect food intake. We also tracked body weight throughout the study (Figure 1D). Body weight was evaluated 32 weeks after exposure to the experimental diets (average age 12.7 mo), and prior to behavioral testing (Figure 1E) and we found significant main effects of genotype (F_(1, 52)_ = 17.60; *p* < 0.0005) and sex (F_(1, 52)_ = 6.44; *p* = 0.014), with the 2n weighing more than 3n animals, and males weighing more than females.

To further evaluate the effect of diet on weight given these group differences, we next assessed the percent change in body weight (Figure 1F,G). Here, there was a significant main effect of diet on percent change in weight (F_(1, 52)_ = 10.57; *p* = 0.002), with ChN animals showing a higher percent change in weight than Ch+ animals. We also found a significant main effect of sex (F_(1, 52)_ = 37.26; *p* < 0.0005), where female mice showed a higher percent change in weight than males. A significant diet by sex interaction (F_(1, 52)_ = 5.32; *p* = 0.025) revealed that males and females were significantly different from each other within diets (ChN *p* < 0.0005, and Ch+ *p* = 0.004), and that female mice gained significantly less weight when fed Ch+ diets when compared to females fed ChN diets (*p* = 0.001). Collectively, this data suggests that dietary Ch+ altered age-related weight gain in females, regardless of equivalent food intake.

### 3.2. Choline Supplementation Does Not Affect Motor Function or Spatial Memory Performance, but Improves Cognitive Flexibility in DS Mice

#### 3.2.1. Rotarod

Choline is a necessary precursor to acetylcholine needed for muscle function, and because we previously documented motor impairments in both control and an AD mouse model with choline deficiency [23], we assessed the effect of diet upon motor function using the rotarod test (Figure 2A; n = 5–9 per sex, diet and genotype). Analysis revealed no significant differences between any group during either the training days (Figure 2B) or probe trial (Figure 2C), indicating that Ts65Dn mice do not exhibit motor impairment as tested by the rotarod, and no motor-specific benefits resulted from choline supplementation at adulthood.

#### 3.2.2. RAWM

Previous work shows marked improvements in spatial memory for trisomic offspring born to maternally supplemented dams [27,29] and in APP/PS1 mice supplemented in adulthood [40]. Here, we tested the ability of Ts65Dn mice to navigate a radial arm water maze (RAWM), which contains a hidden, submerged platform in one arm of the maze, termed the goal arm; when septohippocampal-dependent spatial memory circuits are functioning, animals are able to navigate to the goal arm using external maze cues and use the platform to escape the maze (Figure 2D). We excluded four animals due to their inability to swim, resulting in N = 56 (n = 5–9 animals per sex, diet, and genotype) for this analysis.

One strategy used by mice to find the goal arm is chaining, or the consecutive entry of arms in a circular manner, which does not rely on spatial memory. Here, we found no significant differences between groups in number of chaining events (Figure 2E), suggesting that mice were not using this strategy to find the goal arm. We next evaluated how long it took animals to reach the platform within the goal arm (latency; Figure 2F). We found a significant main effect of genotype (F_(1, 48)_ = 62.35; *p* < 0.0005), where 3n animals took significantly longer to reach the platform across the 6 days of testing. We also found a significant main effect of day (F_(5, 240)_ = 20.48; *p* < 0.0005) with latency declining over time, and a significant genotype by day interaction (F_(5, 240)_ = *p* < 0.0005). Follow-up testing revealed that 3n animals took significantly longer to find the platform on Days 2–6 (*p* < 0.0005 for each). Similarly, the number of trials in which animals found the platform increased over time but showed genotype differences (Figure 2G): We found main effects of genotype (F_(1, 48)_ = 49.15; *p* < 0.0005) and day (F_(5, 240)_ = 11.40; *p* < 0.0005), as well as a significant genotype-by-day interaction (F_(5, 240)_ = 2.78; *p* = 0.018). Follow-up testing revealed that 3n mice found the platform significantly less often than 2n mice on Days 2–6 (*p* < 0.0005 for each). Thus, animals performed better over time as they learned to find the platform, but 2n animals consistently outperformed 3n animals, and there was no effect of Ch+.

To assess arm entry errors, we first determined the average number of arm entries in general (Appendix A) and found a significant main effect of day (F_(5, 240)_ = 3.59; *p* = 0.004), and a significant genotype-by-day interaction (F_(5, 240)_ = 11.3; *p* < 0.0005). Follow-up testing revealed that 3n animals entered significantly fewer arms on average during days 1 and 2 (*p* < 0.0005 and *p* = 0.004, respectively). In trials where more than one arm was entered, we assessed entry errors directly (Appendix A). A significant main effect of day (F_(5, 255)_ = 5.96; *p* < 0.0005) revealed a general decrease in entry errors over time. A significant genotype-by-day interaction (F_(5, 255)_ = 10.29; *p* < 0.0005), when probed in follow-up testing, revealed that 3n animals made more entry errors than 2n animals on day 1 (*p* < 0.0005), day 5 (*p* = 0.001) and day 6 (*p* = 0.005). Similarly, we assessed repeat entry errors in trials where more than one arm was entered (Appendix A). We observed a significant main effect of day (F_(5, 255)_ = 3.3; *p* = 0.007) as repeat entry errors decreased with time, and a significant genotype-by-day interaction (F_(5, 255)_ = 7.4; *p* < 0.0005). Follow-up testing revealed that 3n animals made more repeat entry errors than 2n animals on day 1 (*p* = 0.001) and day 5 (*p* = 0.005). Collectively, low numbers of arm errors and repeat entry errors can be a result of animals making fewer errors on their way to the platform or can be due to low entries into arms in general (see Figure 2D). Thus, to better assess the ability of animals to learn the location of the goal arm across testing, we measured the ratio of correct goal arm entries to total entries, which yields a scale of 0–1 (Figure 2H). We found a significant main effect of genotype (F_(1, 48)_ = 34.62; *p* < 0.0005), with 3n animals achieving lower scores overall. A significant main effect of day (F_(5, 240)_ = 17.5; *p* < 0.0005) revealed that scores improved over time, but a significant genotype-by-day interaction (F_(5, 240)_ = 6.76; *p* < 0.0005)—after follow-up testing—revealed that 3n animals earned lower scores than 2n mice on day 1 (*p* < 0.0005), day 2 (*p* = 0.006), day 4 (*p* < 0.0005), and day 5 (*p* < 0.0005). Thus, we found that 3n mice performed worse in every RAWM metric assessed, with no effects of diet on performance, suggesting that Ch+ diets, when initiated in adulthood, do not affect impairment within this form of spatial learning.

#### 3.2.3. IntelliCage

We have previously demonstrated the utility of the IntelliCage automated behavioral phenotyping system (Figure 3A,B) to explore various behavioral outputs in AD mouse models [51,52]. The IntelliCage uses animals’ motivation to consume water to drive learning within operant corners (Figure 3B) where access to water can be made contingent on learned behaviors. Unfortunately, due to aggression between males when co-housed within the IntelliCage apparatus, four males had to be euthanized early due to severe fight wounds, nine males required treatment by veterinary staff due to mild-to-moderate fight wounds, and all males were returned to their home cages to recover until the end of the study while females proceeded with IntelliCage testing.

Twenty-nine female mice (n = 5–8 per diet and genotype) were tested within the IntelliCage apparatus over the course of 18 days, with 6 days for cage adaptation phases (2 days each for free adaptation, door adaptation, and nosepoke adaptation), 6 days for a place preference task, and 6 days for a reverse place preference task. Animals were removed from IntelliCage testing following failure to drink for 24 h, per institutional protocol. In total, six animals were removed permanently due to failure to drink: n = 3 (2 2n, ChN and 1 3n, Ch+) animals during the adaptation phases, n = 2 (2n, Ch+) animals during the place preference task, and n = 1 (3n, Ch+) during the reverse place preference task. Details of the statistical main effects and interactions for each measure are shown in Table 1, with significant results from follow-up testing of interactions explained below.

During free adaptation, total visits to any corner (Figure 3C) were significantly decreased on day 2. We also found significant effects of day on total licks, a significant effect of genotype on total licks, a significant genotype-by-diet interaction, and a significant diet-by-day interaction (Figure 3D). Follow-up analyses revealed that Ch+ animals made more licks on day 2 (*p* = 0.002), Ch+ 2n animals licked less than Ch+ 3n animals (*p* = 0.009), and 3n mice on the Ch+ diet licked more than 3n ChN mice (*p* = 0.001). During door adaptation, we found a significant genotype effect in total visits (Figure 3E), where 3n mice made more total visits than 2n mice. For total licks (Figure 3F), we found a significant effect of day, a genotype-by-day interaction, a diet-by-day interaction, and a genotype-by-diet-by-day interaction; follow-up testing revealed that 2n mice licked less than 3n mice on day 1 (*p* = 0.004) and ChN mice trended towards fewer licks than Ch+ mice on day 1 (*p* = 0.007, which was not significant following a multiple comparisons correction threshold of *p* < 0.00625). Nosepoke adaptation requires animals to learn to nosepoke to gain water access. We found significant effects of day for total visits (Figure 3G), corner visits with >1 lick (Figure 3H), and total licks (Figure 3I). Follow-up testing for significant interactions within corner visits with >1 lick revealed that within Ch+ animals, 3n animals made more corner visits with >1 lick than 2n on day 2 (*p* = 0.009). Collectively, the data from adaptation phases suggest that animals were able to learn where and how to access water, that exploratory behavior of the novel environment decreased over time, and that 3n mice made more visits, consistent with observed hyperactivity in other studies [56,57,58]. Furthermore, the data revealed some differences in water consumption based on diet, with Ch+ animals drinking more at times than ChN animals.

To complement our examination of septohippocampal-dependent spatial learning, animals underwent a place preference task, where access to water was restricted for each mouse to one assigned corner such that they were required to use spatial cues to locate the correct corner for water access. While the number of visits overall to any corner (Figure 4A), total licks (Figure 4B), visits to the assigned corner (Appendix A), and visits to the assigned corner with >1 lick (Figure 4C) were not significantly different over time, the percent of visits to the assigned corner (Appendix A) and the percent of visits to the assigned corner with >1 lick (Figure 4D) showed increases over time, consistent with animals making more visits to the assigned corner to lick as they learned. Additionally, follow-up testing of a significant diet-by-day interaction in total licks showed that Ch+ animals licked more than ChN animals specifically on day 6 (*p* = 0.001). No genotype differences were observed, contrasting with our RAWM data and suggesting that 3n females perform better in the IntelliCage than in the RAWM. This is consistent with our previous results suggesting that female mouse models of AD perform better in the IntelliCage than in the Morris Water Maze, another spatial memory paradigm [52]. It is also consistent with a previous report on place learning in the IntelliCage for female Ts65Dn mice [59].

Lastly, animals underwent a reverse place preference task, where the assigned corner from the previous task no longer allowed water access, instead requiring animals to switch to the corner on the opposite side of the apparatus for water, indicating behavioral flexibility, which is mediated by the prefrontal cortex and hippocampus. Intriguingly, while total visits to any corner (Figure 4E) were relatively stable across time, total licks showed a significant day effect and a significant diet-by-day interaction; on day 1, the Ch+ animals licked significantly more (*p* < 0.0005) in their new assigned corner (Figure 4F). Similarly, we found significant diet-by-day interactions in the number of visits to the newly assigned corner (Appendix A) and number of visits to the newly-assigned corner with >1 lick (Figure 4G); follow up testing revealed that Ch+ animals made significantly more visits to the newly-assigned corner on day 1 (*p* = 0.001), and more visits with >1 lick on day 1 (*p* < 0.0005). The percent of visits to the assigned corner (Appendix A) and percent of visits with >1 lick (Figure 4H) did not show any differences across groups, and while there was a significant genotype by day interaction in the latter, it was not significant when follow-up testing was corrected for multiple comparisons. Lastly, we looked at the percent of visits animals made to their previously assigned corner from the place preference task (Figure 4I) on the first day of the reversal task and found that Ch+ animals made significantly fewer visits to their old corner. Collectively, these data suggest that a Ch+ diet in adulthood plays a role in modifying animals’ cognitive flexibility in adapting to a new corner for water access, consistent with previous work showing improvements to prefrontal cortical behaviors in trisomic offspring of MCS treated Ts65Dn mice [30,31,32].

### 3.3. Choline Supplementation Lowers Fasting Glucose and Improves Peripheral Inflammation

For all analyses following the divergence in treatment between females and males (separated due to aggression in the IntelliCage), subsequent data from sexes were analyzed independently.

#### 3.3.1. Glucose Metabolism

Diabetes incidence in DS is high, particularly in relation to obesity [60,61,62]. To understand the effect of adulthood choline supplementation on glucose metabolism, at approximately 14.2 mo of age (1 week prior to euthanasia), we fasted mice overnight for at least 16 h and tested both fasting glucose and glucose tolerance (n = 5–9 per sex, diet, and genotype). The analysis of glucose levels across time (Figure 5A) revealed a significant main effect of time for both males (F_(5, 110)_ = 47.41; *p* < 0.0005) and females (F_(5, 125)_ = 53.95; *p* < 0.0005); in males there was a trending main effect of diet (F_(1, 22)_ = 3.34; *p* = 0.081) and a trending diet-by-timepoint interaction (F_(5, 110)_ = 2.0; *p* = 0.084). To further assess the animals’ ability to respond to the glucose challenge over time, we quantified the area under the curve (Figure 5B) and found no significant effects of diet or genotype in either sex. In males, however, there was a significant main effect of diet (F_(1, 22)_ = 6.14; *p* = 0.021) on fasting glucose (Figure 5C), with Ch+ males demonstrating lower fasting glucose levels. Females did not show this effect (F_(1, 25)_ = 1.84; *p* = 0.187). Thus, Ch+ in adulthood may aid in lowering fasting glucose, and modestly affects the ability of an animal to accommodate the increased glucose with adequate metabolic response.

#### 3.3.2. End Body Weight

When animals were euthanized at an average age of 14.3mo (n = 5–8 per sex, diet, and genotype), we measured endpoint body weight (Figure 5D). In males, we found a significant main effect of genotype on body weight (F_(1, 25)_ = 33.22; *p* < 0.0005), with 3n males significantly smaller than 2n males. Females also showed a significant main effect of genotype on body weight (F_(1,25)_ = 6.45; *p* = 0.018) and a significant genotype-by-diet interaction (F_(1, 25)_ = 4.59; *p* = 0.042); follow-up testing revealed that 3n females were significantly smaller than 2n females specifically within the ChN group (*p* = 0.003). Previous work has shown reduced body size in 3n animals versus their 2n littermates at 1–2 mo [63,64] and 15 mo [29]. Previously, we found no genotype differences in weight up to 12 months of age, highlighting the variability of morphological features in Ts65Dn mice [47,65].

#### 3.3.3. Circulating Choline

We have previously shown that fasting plasma choline decreases with age in mouse models of AD and that choline deprivation also reduces circulating choline levels [23,24]. Here, to determine circulating total choline in Ts65Dn mice both generally and with choline supplementation, we measured both baseline and endpoint fasting levels from a subset of animals (Figure 5E; n = 3 per sex, diet, and genotype), and found a significant main effect of timepoint in both males (F_(1, 8)_ = 23.46; *p* = 0.001) and females (F_(1, 8)_ = 24.19; *p* = 0.001), but no main effects of diet or genotype. This suggests that circulating choline declines with age in the Ts65Dn mouse model, consistent with other mouse models, but supplementation with exogenous choline does not change this decline.

#### 3.3.4. Hepatic Steatosis

Inadequate choline intake leads to hepatic steatosis and liver damage [19,66], but hepatic steatosis has also been observed in Ts65Dn mice fed high-fat diets [67], and is associated with excess weight in children and adolescents with DS [68]. Whether choline supplementation can improve liver health in DS has yet to be explored. Thus, we analyzed liver tissue from a subset of animals for microvesicular steatosis, macrovesicular steatosis, and hepatocellular hypertrophy (Appendix A) via methods described in mice previously and as used previously in our lab [23,54]. Sections of livers from a subset of animals (n = 4 per sex, diet, and genotype) were stained with hematoxylin and eosin (H&E) and scored for steatosis and hypertrophy by two independent observers blinded to animal groups. Composite scores were highly variable across animals of both sexes regardless of diet and genotype (Appendix A), consistent with previous findings [67]. However, there was a moderate but significant positive correlation (r_(31)_ = 0.4786; *p* = 0.0056) between the steatosis score and the percent weight change by 32 weeks (Appendix A). Thus, Ch+ does not appear to deter the development of steatosis in animals with higher body weights.

#### 3.3.5. Peripheral Inflammation

Elevated peripheral inflammation is observed in DS, and cytokines such as interleukins (IL), tumor necrosis factor α (TNF-α), and components of interferon signaling have been documented as increased in blood samples from individuals with DS [15]. To determine whether adult choline supplementation altered peripheral cytokine levels, we evaluated plasma levels of 23 cytokines in a subset of animals (n = 3–5 per sex, diet, and genotype) using the Bio-Plex^®^ suspension array multiplexing system [53]. We found no significant differences in cytokine levels across groups in males (Figure 6A; Table 2). However, female 3n mice showed significant elevations in 17 of the 23 cytokines (Figure 6B; Table 2; Appendix A–H). Notably, Ch+ diets significantly decreased 10 of the 23 cytokines, many of which are pro-inflammatory: Granulocyte-macrophage colony-stimulating factor (GM-CSF; Figure 6C), interferon γ (IFN-γ; Figure 6D), interleukins IL-1α, IL-1β, IL-3, IL-6, IL-9, IL-12p70, IL-17 (Figure 6E–K), and TNF-α (Figure 6L). This is consistent with previous reports showing choline’s ability to attenuate systemic inflammation [69,70].

### 3.4. Choline Supplementation in Adulthood Does Not Alter Basal Forebrain Cholinergic Neuron Loss in DS Mice

Much of the previous literature involving perinatal choline supplementation in Ts65Dn mice investigated its effects on BFCNs and BFCN-modulated behaviors [27,28,29,30,31,32]. Here, we sought to determine whether adult choline supplementation results in a similar benefit. We stained sections from a subset of animals (n = 3–4 per sex, genotype, and diet; Figure 7A) for choline acetyltransferase (ChAT), and used the optimized optical fractionator method for counting the number of ChAT+ cells within the combined MS/VDB (Figure 7B); these regions are contiguous and difficult to outline independently, and previous work found that there is no difference between analyzing these regions separately vs. combining the MS/VDB [31]. We found no significant differences in cell number between genotypes or diet (Figure 7C) for either sex. Female 3n mice had a significantly larger MS/VDB area (F_(1, 11)_ = 19.47; *p* = 0.001; Figure 7D), whereas males did not show differences in area across genotype or diet. Neither sex showed genotype or diet differences for cell density (Figure 7E). To confirm that genotype-specific effects in MS/VDB cell density were aligned with previous studies, we combined data sets for male and female mice, despite their unequal treatment after the IntelliCage testing. We found a trend (F_(1, 22)_ = 19.18; *p* = 0.05) towards a decrease in cell density in 3n animals compared to 2n animals (Figure 7F), consistent with previous studies and suggesting that a Ch+ diet in adulthood does not alter the MS/VDB as observed with MCS [27,28].

## 4. Discussion

Here, we tested the effects of dietary choline supplementation in adulthood using the Ts65Dn mouse model of DS. Previous studies in mice have demonstrated that BFCNs—and behavioral outcomes dependent on these neurons—benefit from perinatal choline supplementation, and MCS in humans provides cognitive advantages during early childhood development, particularly in tests of attention [71,72]. Thus, there is evidence that perinatal supplemental choline benefits development, with lasting outcomes in mice and humans—both generally and in DS specifically [73,74]. Although studies from 40 years ago first suggested no cognitive benefits of supplemental choline in young adults [75] and in older adults [76], these studies examined short-term adulthood choline supplementation. More recently, evidence has suggested that higher choline intake (from a combination of diet and supplementation) is associated with better performance on cognitive tests [77]. Notably, postmenopausal women have higher dietary needs for choline than do premenopausal women, due to lower estrogen concentrations, but this is not currently considered in the recommended AI for choline [78]; this is despite the fact that postmenopausal women with low choline intake are more likely to develop hepatic fibrosis [79]. We found that adulthood choline supplementation in Ts65Dn mice modestly improved fasting glucose in male mice and lowered age-related weight gain and inflammation in females, and that it improved cognitive flexibility in females. This suggests that, while early-life choline supplementation is important, later supplementation can still influence key parameters important for general health outcomes, which—when dysfunctional—are also risk factors for Alzheimer’s disease.

Simple diet modifications that lead to clinically meaningful outcomes may be especially valuable for individuals with DS. Children and adolescents with DS have higher rates of obesity, and obesity in DS is positively associated with dyslipidemia, hyperinsulinemia, and other factors [80], though overall cardiometabolic risk was similar to that of adults generally [81]. One study of 9917 DS and 38,266 control individuals suggests that diabetes incidence in DS is four times higher [60], and the relationship between obesity and T2D risk in DS appears especially strong at younger ages [62,82]. In an additional study, obese, overweight female DS individuals demonstrated the highest measures of insulin dysfunction [61]. Furthermore, another condition related to increased obesity in DS is hepatic steatosis, with high prevalence of nonalcoholic fatty liver disease (NAFLD) observed in DS children and teenagers despite no other known causes for liver disease apart from excess weight, which suggests that DS itself may predispose individuals to fatty liver disease irrespective of body mass index [68]. Given that these clinical factors are also known risk factors for the development of AD in the general population [9,10], it is essential to understand how to mitigate these risks in the already AD-vulnerable DS population.

Our data suggest that adulthood choline supplementation reduced weight gain particularly in female mice—a finding not related to MCS [83], and one which occurred despite the lack of differences in Ch+ versus ChN food intake. Whether supplemental choline during adulthood altered weight gain by changing metabolism and/or by increasing animals’ activity remains to be determined. Trisomic mice have been shown to have elevated locomotor activity and increased energy expenditures when compared to disomic littermates, but still develop dyslipidemia and variable levels of steatosis when fed high-fat diets [67]. Here, our IntelliCage data suggest that 3n females had higher levels of physical activity, as evident in measures such as total corner visits. However, there were no effects of diet on total corner visits, suggesting that Ch+ animals were not more physically active. Although hepatic steatosis was variable and not affected by Ch+ diet, we found that steatosis correlated with the amount of weight gained over time. Thus, controlling weight gain in individuals coupled with additional choline intake may be a potential strategy to address the high incidence of NAFLD in DS, given that its incidence is a potential risk factor for AD [84,85]. In male mice, while we did not see a pronounced diet effect on weight gain, Ch+ significantly lowered fasting glucose regardless of genotype and trended towards a more rapid response to glucose challenge, though the latter was not significant. Previous work has identified the triplicated DS protein regulator of calcineurin 1 (RCAN1) as a driver of T2D in Ts65Dn mice and in humans with T2D, with a loss of RCAN1 gene methylation leading to its increased expression in pancreatic islets [86]. Whether supplemental choline was able to alter the methylation status of RCAN1 was not addressed here, but further research is warranted to understand the utility of choline supplementation in attenuating glucose dysregulation.

Interestingly, we found that administration of the Ch+ diet throughout adulthood of Ts65Dn mice did not prevent age-related decline in circulating total choline. One possible explanation is that our measurement of fasting choline levels may have depleted any circulating choline from the plasma such that an effect of supplementation is masked. However, it is also possible that circulating total choline does not reflect the partitioning of choline across different organ systems altered with supplemental choline. Previous work administering deuterium-labeled choline to Ts65Dn mice has shown that choline metabolism is altered in 3n mice and that, in general, choline is preferentially partitioned to the brain [83]—a finding also supported in choline deficient mice [87]. It was also demonstrated in Ts65Dn mice that offspring of MCS dams experienced a permanent enhancement of choline uptake and metabolism, primarily by effects on endogenous production of choline but also with preferential partitioning of choline towards the brain [83]. In humans, choline uptake into the brain also decreases with age [88], and phosphatidylcholine—one of the most common membrane phospholipids in the brain—decreases in the prefrontal cortex of adults with aging [89]. Further, higher plasma phosphatidylcholine levels were associated with better performance in elderly adults on tasks measuring prefrontal-cortex mediated cognitive flexibility [90]. Notably, we found that Ch+ mice were better than ChN mice at adapting to the reversal phase of the place preference task, suggesting enhanced cognitive flexibility. Reversal learning in spatial paradigms is governed by the orbitofrontal cortex in mice (reviewed previously [91,92]). While the nature of basal forebrain–frontal cortex innervation is a current topic of study, evidence points to cholinergic innervation of the rodent orbitofrontal cortex by the NBM [93] as well as reciprocal innervation between cholinergic neurons, such as those in the horizontal diagonal band (HDB), and the orbitofrontal cortex [94,95]. Past work has shown that attentional performance in 3n mice—another behavior modality relying on the prefrontal cortex—benefits from MCS [30,31,32] and that this performance correlates with BFCN density in the NBM/substantia innominata (SI) [31]. Whether the observed effect of Ch+ on cognitive flexibility here was due to changes in BFCNs outside of the MS/VDB and/or to changes in the frontal cortex remains to be determined. Furthermore, whether Ch+ changed choline metabolism and/or altered the levels of phosphatidylcholine in the bloodstream or frontal cortex was not addressed in this study but may be worthwhile to pursue in the future.

Our data also suggest that Ch+ in adulthood failed to ameliorate deficits in spatial learning in the RAWM; similarly, we saw no changes to the MS/VDB of trisomic animals as a result of Ch+ in adulthood. That adulthood Ch+ failed to attenuate RAWM performance decrements in 3n animals suggests that the benefits of choline supplementation on this behavior modality and its related neurocircuitry are largely developmental. However, the mechanisms by which perinatal choline benefits basal forebrain circuitry remain unclear. MCS has previously been shown to improve RAWM performance which in turn correlates with MS cell density [27], hippocampal ChAT innervation [96], and neurogenesis markers [29]. Two previous MCS studies have shown a rescue of BFCN number or density in the MS/VDB within 3n offspring of MCS dams at similar ages to ours [28,42]. And while previous reports investigating Ts65Dn mice of similar ages have showed discrepancies in 2n vs. 3n animals in terms of MS/VDB cell number and density [27,28,31,97], our analysis of BFCNs in the MS/VDB did not return differences between 2n and 3n animals in regards to cell numbers or density, though we did show in 3n females an enlarged area as reported previously in both male and female 3n mice of similar ages [27]. One limitation of this work is that, after dividing into sexes, the number of animals used for analysis of BFCNs in the MS/VDB was low in comparison to previous work [27,28]. The variability in BFCN numbers and density in this strain is high, as reflected by other groups who have failed to find genotype differences in cell number or density within the MS/VDB at a similar age in Ts65Dn mice [98], or have found that the density of ChAT+ neurons in regions such as the NBM/SI are not different between 2n vs. 3n animals and do not change with MCS, despite correlating with attentional performance [31]. Thus, focusing solely on ChAT+ cell numbers or density may not adequately illustrate how choline supplementation alters hippocampal- and cortex-dependent behaviors modulated by the basal forebrain. To this end, others have begun looking at additional basal forebrain cell types, such as parvalbumin+ inhibitory neurons [28], as well as investigating changes in gene expression that occur as a result of MCS in specific brain regions and neuron populations, including MS/VDB BFCNs [33], CA1 hippocampal neurons [34,35], and frontal cortical bulk tissue [36].

A particularly intriguing finding in this work is that adult choline supplementation lowered peripheral inflammation in female mice. Adults with DS display global immune dysregulation [99], and proteomic evaluation of blood samples from individuals with DS shows elevated peripheral IFN-y signaling, TNF-α, and IL-6 [15]. Further, CD4+ T-cells from DS individuals are polarized towards Th1/Th17 responses, higher basal levels of IFN signaling, and hyperresponsiveness to IFN stimulation [100]. However, while peripheral inflammation has been linked to obesity in Ts65Dn mice [101], little is known about age-related changes in peripheral inflammation in Ts65Dn mice or how dietary interventions alter peripheral inflammation in this mouse model. Our results indicate that Ch+ in female mice reduced several cytokines relevant to DS co-morbidities. For example, IFN-γ, IL-6, and TNF-α were all upregulated in 3n female mice, similar to in DS individuals [15], as was IL-12p70—the interleukin upstream of the release of IFN-γ and TNF-α—and all of these were reduced by the Ch+ diet. IL-17 was also increased in 3n female animals, consistent with elevated Th17 response in humans [100], and was also decreased by supplemental choline. Current studies are exploring Janus kinase (JAK) inhibition in autoimmune disorders of DS [102]; several cytokines that act as upstream regulators of JAK signaling were decreased by choline supplementation in this study. Interestingly, six of the cytokines elevated in 3n mice that were attenuated with the Ch+ diet (IL-1α, IL-1β, IFN-γ, TNF-α, IL-3, and IL-6) have been associated with the development of insulin resistance in T2D [103,104,105]. Lastly, IFN-γ can enter the brain and promote detrimental neurodegenerative outcomes [106], while pro-inflammatory cytokines TNF-α, IL-6, and IL-1β have also been linked to AD [11]. Further studies should investigate whether supplemental choline can similarly attenuate aberrant inflammatory cascades in individuals with DS and AD.

## 5. Conclusions

This work adds to the existing body of data on choline supplementation by demonstrating that the timing of choline supplementation offers different benefits, which may also differ between sexes. Past work has supported the value of perinatal supplementation, which is crucial given the low intake levels of choline observed worldwide and the ability of early supplementation to prevent a trajectory of pathological outcomes. This current study suggests that adulthood supplementation can still be beneficial, particularly in individuals with DS, through its ability to reduce inflammation, improve cognitive flexibility, and affect metabolic parameters such as obesity and glucose dysregulation that are themselves risk factors for AD.

## Figures and Tables

**Figure 1 nutrients-16-04167-f001:**
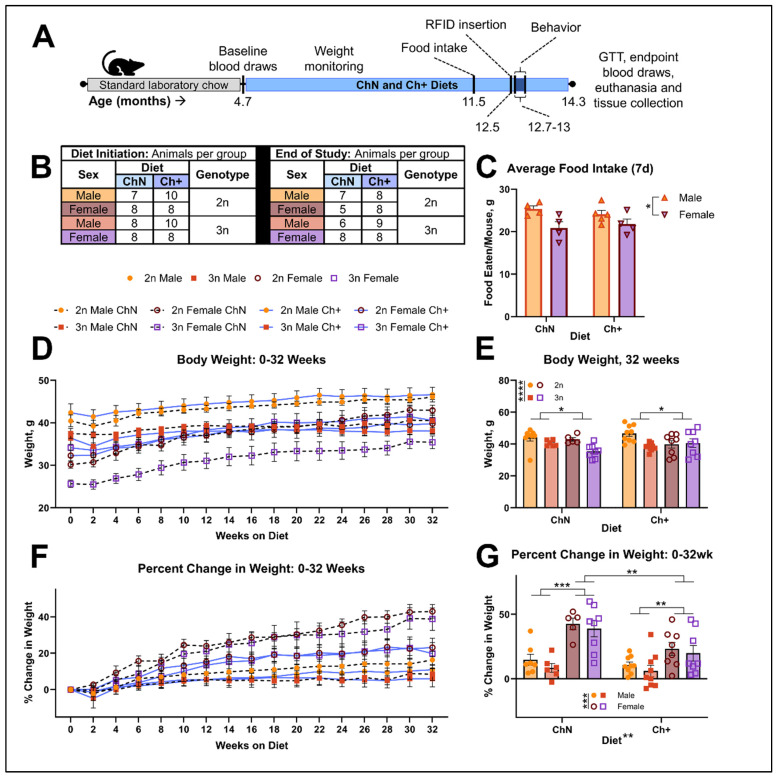
Study parameters, weight gain, and food intake. (**A**) Study timeline. (**B**) Numbers of animals per genotype (2n = disomic, 3n = trisomic), diet (ChN = choline normal, Ch+ = choline supplemented), and sex. (**C**) Average food intake per cage (n = 4–5 per sex and diet, adjusted for weight of mice per cage) after 28 weeks on experimental diets. (**D**) Raw body weights in grams (g) across the study from baseline to 32 weeks on experimental diets (n = 5–9 per sex, diet, and genotype). (**E**) Quantification of average body weight at 32 weeks on experimental diets, prior to behavior. (**F**) Percent change in body weights across the study from baseline to 32 weeks on experimental diets. (**G**) Quantification of average percent change in body weight at 32 weeks on experimental diets. Error bars represent SEM; * *p* < 0.05, ** *p* ≤ 0.01, *** *p* ≤ 0.001, **** *p* ≤ 0.0001 (multivariate analysis of variance; MANOVA).

**Figure 2 nutrients-16-04167-f002:**
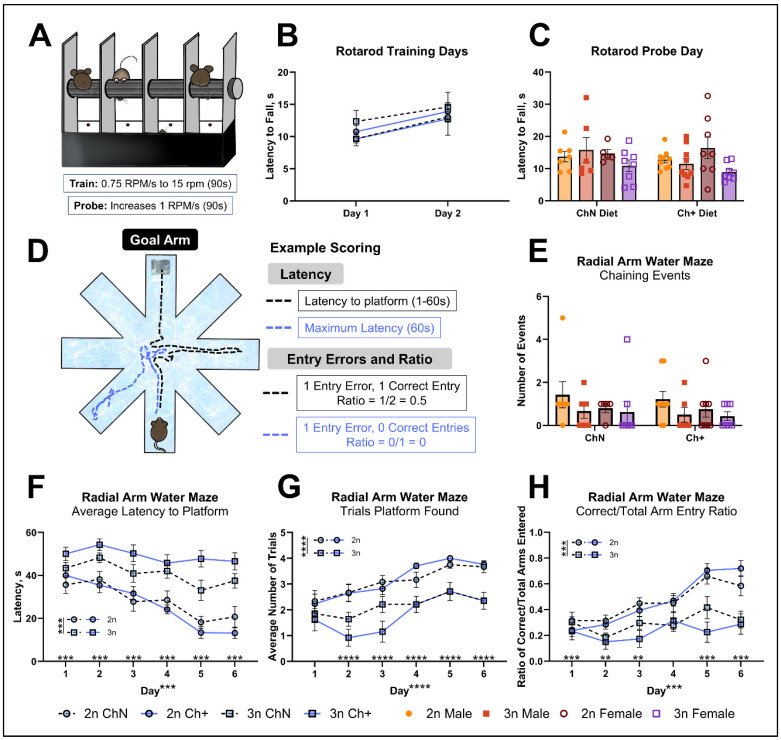
Rotarod and radial arm water maze testing. (**A**) The rotarod was used to test animals’ motor function (n = 5–9 per sex, diet, and genotype). (**B**,**C**) Latency to fall from the rotarod on the training days (**B**) and probe day (**C**). (**D**) The radial arm water maze was used to test animals’ spatial learning and memory (n = 5–9 animals per sex, diet, and genotype), and scored for their latency to reach the platform as well as arm entries. (**E**) Average number of chaining events. (**F**) Latency to reach platform. (**G**) The number of trials where a platform was found within 60 s. (**H**) The ratio of successful trials to total arm entries. For graphs (**B**,**F**–**H**), sexes are collapsed for simplicity as no main effect of sex was detected. Error bars represent SEM; ** *p* ≤ 0.01, *** *p* ≤ 0.001, **** *p* ≤ 0.0005 (MANOVA).

**Figure 3 nutrients-16-04167-f003:**
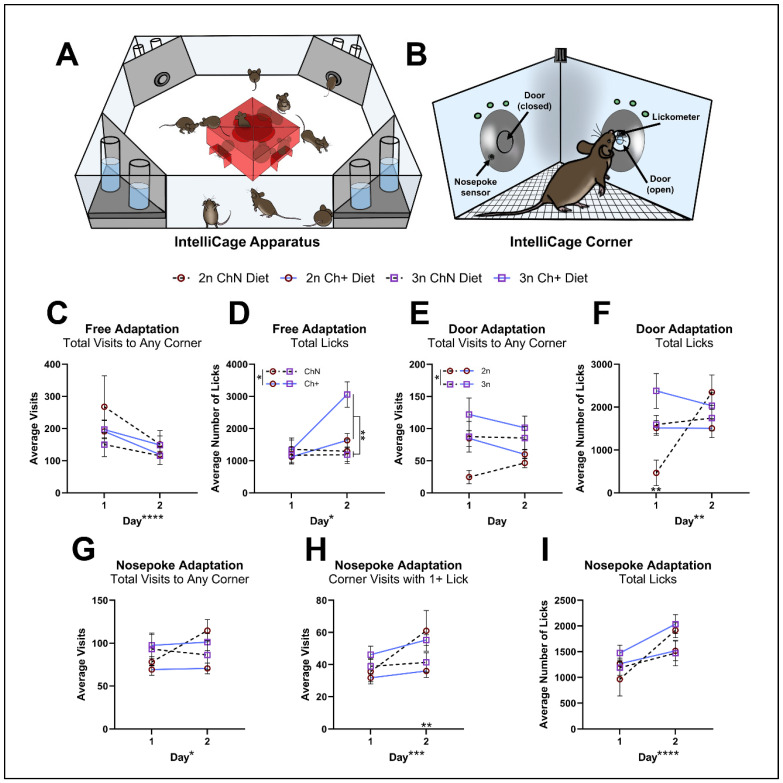
The IntelliCage apparatus and adaptation phases. (**A**) The IntelliCage apparatus was used to explore additional behavior outputs. Female mice only (n = 5–8 per diet and genotype) were tested due to aggression between males when co-housed. Animals were tested over 18 days, with 6 days for adaptation phases, a place preference task, and a place preference reversal task, respectively. Water was accessible in operant corners (**B**) 24 h/day, but after the adaptation phases, water was only available to each mouse in a single assigned corner. A total of n = 6 animals were removed from the IntelliCage due to failure to drink (2 each of 2n ChN, 2n Ch+ and 3n Ch+). (**C**,**D**) Animals were assessed for total visits (**C**) and total licks (**D**) during the free adaptation phase where animals grow accustomed to the cage. (**E**,**F**) Animals were assessed for total visits (**E**) and total licks (**F**) during the door adaptation phase, where animals are introduced to the motorized door opening upon corner entry. (**G**–**I**) Animals were assessed for total visits (**G**), corner visits with at least 1 lick (**H**), and total licks (**I**) during the nosepoke adaptation phase, where animals learned to nosepoke the sensor in order to receive water. Error bars represent SEM; * *p* < 0.05, ** *p* ≤ 0.01, *** *p* ≤ 0.001, **** *p* ≤ 0.0005 (MANOVA).

**Figure 4 nutrients-16-04167-f004:**
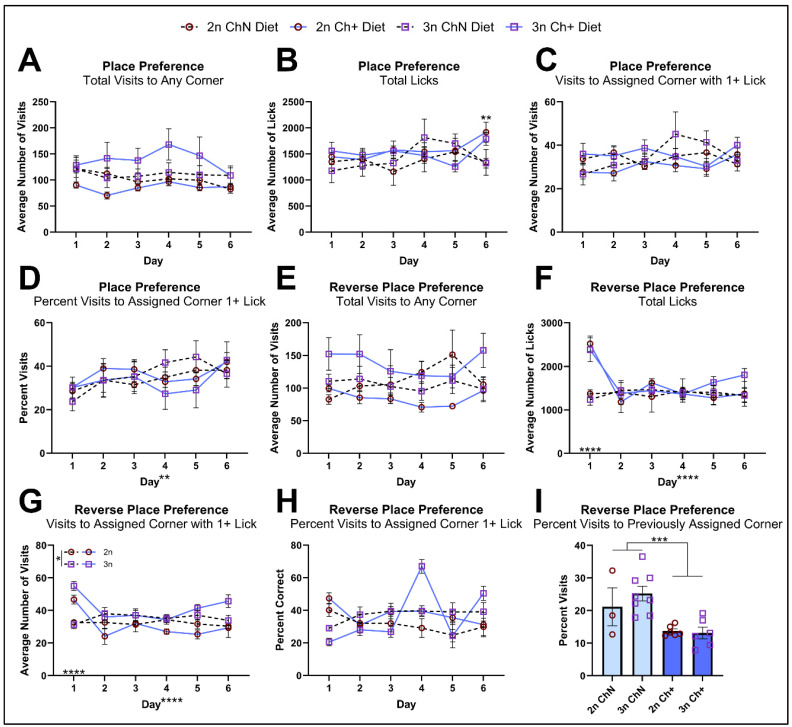
Place preference and reversal learning in the IntelliCage. (**A**–**D**) The place preference task was used to assess spatial learning and memory, as animals were randomly assigned to a corner and only able to consume water in that corner, relying on spatial cues to navigate successfully to that corner (n = 3–8 per diet and genotype). Animals were assed for total visits to any corner (**A**), total licks (**B**), number of visits to the assigned corner with at least 1 lick (**C**), and the percent of visits to the assigned corner with at least 1 lick (**D**). (**E**–**I**) The place preference reversal task was used to assess cognitive flexibility, as the animals’ assigned corners were switched to the corner on the opposite side of the cage. Animals were assessed for total visits to any corner (**E**), total licks (**F**), number of visits to the assigned corner with at least 1 lick (**G**), percent of visits to the assigned corner with at least 1 lick (**H**), and the percent of visits animals made to their previously assigned corner on day 1 (**I**). Error bars represent SEM; * *p* < 0.05, ** *p* ≤ 0.01, *** *p* ≤ 0.001, **** *p* ≤ 0.0005 (MANOVA).

**Figure 5 nutrients-16-04167-f005:**
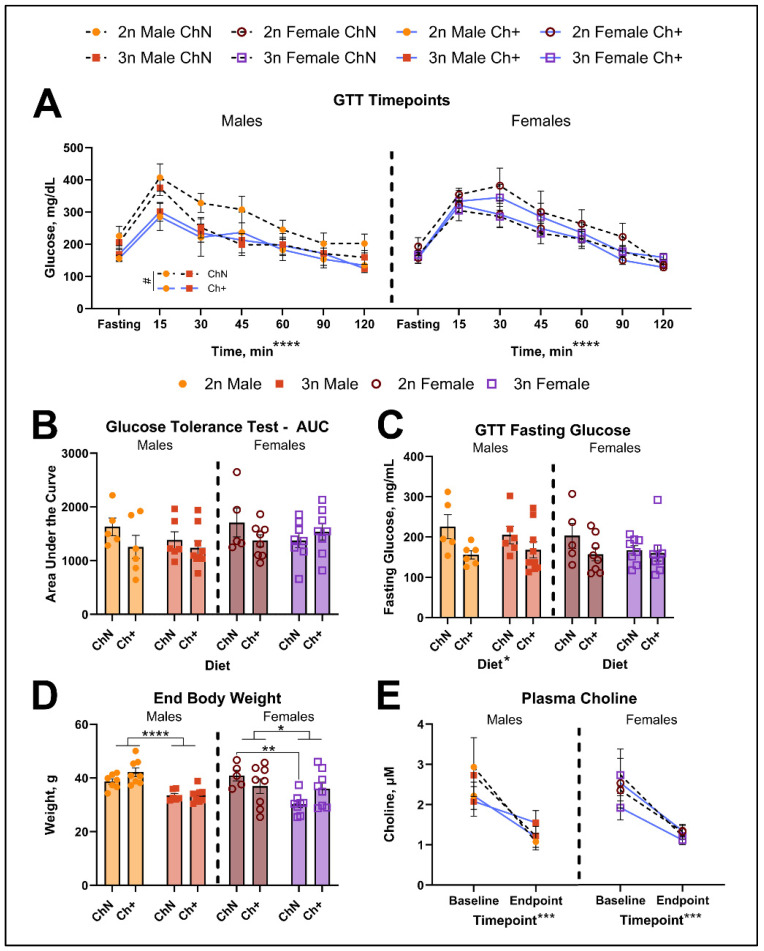
Glucose metabolism, ending body weights, and circulating total choline. (**A**–**C**) Glucose tolerance testing (GTT) was performed to assess glucose homeostasis (n = 5–9 per diet and genotype; sexes analyzed separately). We assessed blood glucose over time (**A**), quantification of area under the curve (AUC; **B**), and fasting glucose (**C**). (**D**) Body weights were measured at end of study (n = 5–8 per diet and genotype, sexes analyzed separately). (**E**) Fasting plasma total choline was measured (n = 3 per diet and genotype, sexes analyzed separately) at baseline and end of study. Error bars represent SEM; * *p* < 0.05, ** *p* ≤ 0.01, *** *p* ≤ 0.001, **** *p* ≤ 0.0005, # is trending (*p* ≥ 0.05 but <0.10) (MANOVA).

**Figure 6 nutrients-16-04167-f006:**
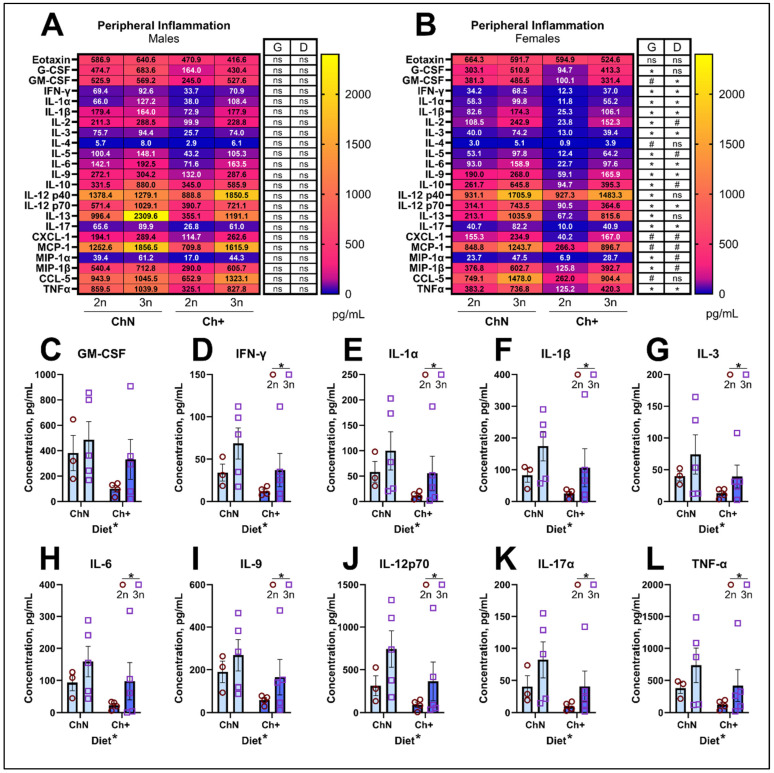
Endpoint peripheral cytokines. (**A**,**B**) Heatmaps demonstrating quantification of peripheral cytokines in blood plasma using the Bio-Plex^®^ suspension array multiplexing system (n = 3–5 per diet (D) and genotype (G); sexes analyzed separately) in male mice (**A**) and female mice (**B**). (**C**–**L**) Individual cytokines that were significantly impacted by diet in female mice included granulocyte-macrophage colony-stimulating factor (GM-CSF; **C**), interferon γ (IFN-γ; **D**), interleukin (IL)-1α (**E**), IL-1β (**F**), IL-3 (**G**), IL-6 (**H**), IL-9 (**I**), IL-12p70 (**J**), IL-17 (**K**), and tumor necrosis factor α (TNF-α; **L**). Error bars represent SEM; * *p* < 0.05, # is trending (*p* ≥ 0.05 but <0.10), ns = non-significant (MANOVA).

**Figure 7 nutrients-16-04167-f007:**
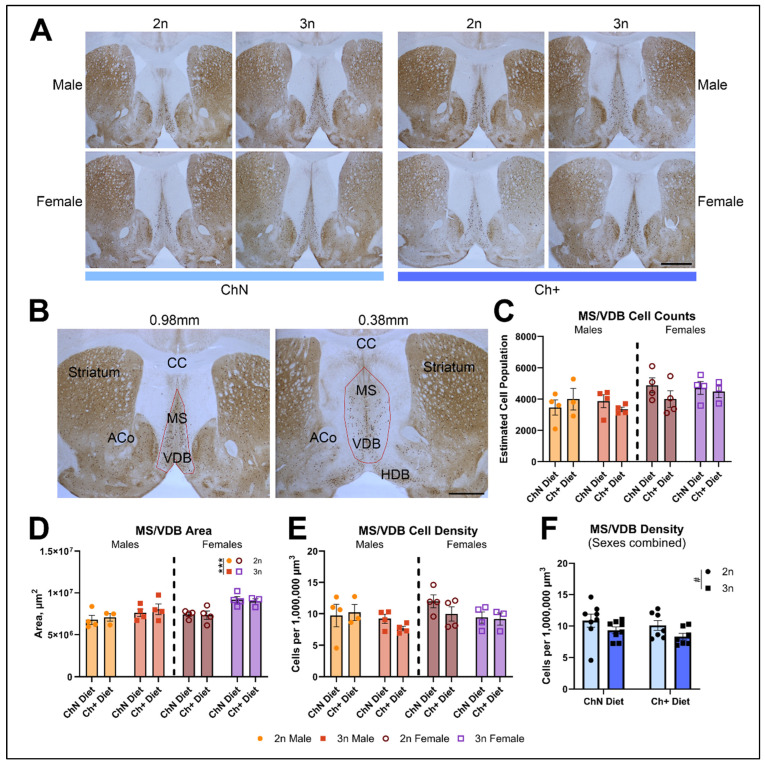
Unbiased stereology of cholinergic neurons in the MS/VDB of the basal forebrain. (**A**) Representative low-magnification photomicrographs of the anterior-most section of the MS/VDB used for unbiased stereology of ChAT+ basal forebrain neurons across all groups (n = 3–4 per diet and genotype; sexes analyzed separately). Scale bars = 1000 μm. (**B**) Low-magnification photomicrographs of the anterior and posterior sections of the MS/VDB (red outline) analyzed via stereology, with approximate coordinates relative to the bregma as well as other anatomical landmarks including the corpus callosum (CC), striatum, anterior commissure (ACo), and horizontal diagonal band (HDB). We assessed cell counts (**C**), region area (**D**), and cell density (**E**), as well as cell densities for sexes combined in order to better compare to the previous literature (**F**; # *p* = 0.05). Error bars represent SEM; *** *p* ≤ 0.001 (MANOVA).

**Table 1 nutrients-16-04167-t001:** IntelliCage Results. The IntelliCage apparatus was used to explore additional behavior outputs over 18 days, with 6 days for cage adaptation, a place preference task (6d), and a place preference reversal task (6d). Female mice only (n = 5–8 per diet and genotype) were tested due to aggression between males when co-housed. A total of n = 6 animals were removed from the IntelliCage due to failure to drink (2 each of 2n ChN, 2n Ch+ and 3n Ch+). Main effects (ME) of genotype (Geno) and diet, as well as Geno x (x used to indicate by in Table 1) Diet interactions, are reported here following MANOVA for each parameter. For parameters measured over several days, ME of day, as well as Geno x Day, Diet x Day, and Geno x Diet x Day interactions are also reported. Significant results from follow-up testing are reported in the manuscript. *p* values of ME and interactions below the threshold for significance (*p* < 0.05) are shown in bold: * *p* < 0.05, ** *p* ≤ 0.01, *** *p* ≤ 0.001, **** *p* ≤ 0.0005.

**ADAPTATION**														
	**ME Geno**		**ME Diet**		**Geno x Diet**		**ME Day**		**Geno x Day**		**Diet x Day**		**Geno x Diet x Day**	
	**F_(1, 22)_**	** *p* **	**F_(1, 22)_**	** *p* **	**F_(1, 22)_**	** *p* **	**F_(1, 22)_**	** *p* **	**F_(1, 22)_**	** *p* **	**F_(1, 22)_**	** *p* **	**F_(1, 22)_**	** *p* **
**Free Adaptation**														
**Total Visits**	0.74	0.4	0.04	0.835	1.89	0.183	22.98	****** <0.0005**	3.51	0.074	0.31	0.583	1.03	0.321
**Total Licks**	2.2	0.152	5.62	*** 0.027**	4.53	*** 0.045**	5.43	*** 0.029**	1.88	0.184	5.91	*** 0.024**	1.47	0.239
**Door Adaptation**														
**Total Visits**	5.36	**0.030**	2.54	0.125	0.09	0.768	0.93	0.344	0.57	0.457	6.26	*** 0.020**	1.13	0.298
**Total Licks**	2.81	0.108	1.25	0.276	0.58	0.456	10.29	**** 0.004**	15.52	**0.001**	20.72	****** <0.0005**	7.08	*** 0.014**
**Nosepoke Adaptation**														
**Total Visits**	0.64	0.433	0.35	0.560	1.63	0.215	5.62	*** 0.027**	7.84	**0.010**	2.82	0.107	9.72	**** 0.005**
**Corner Visits 1+ Lick**	0. 79	0.384	0.16	0.694	6.66	**0.017**	14.42	***** 0.001**	2.73	0.113	1.77	0.197	6.52	*** 0.018**
**Total Licks**	0.44	0.515	0.97	0.336	1.55	0.226	35.17	****** <0.0005**	1.06	0.315	1.46	0.240	7.99	**** 0.010**
**PLACE PREFERENCE**														
	**ME Geno**		**ME Diet**		**Geno x Diet**		**ME Day**		**Geno x Day**		**Diet x Day**		**Geno x Diet x Day**	
	**F_(1, 20)_**	** *p* **	**F_(1, 20)_**	** *p* **	**F_(1, 20)_**	** *p* **	**F_(5, 100)_**	** *p* **	**F_(5, 100)_**	** *p* **	**F_(5, 100)_**	** *p* **	**F_(5, 100)_**	** *p* **
**Total Visits**	2.67	0.118	0.09	0.773	1.39	0.251	1.62	0.161	0.41	0.840	0.93	0.463	1.19	0.318
**Total Licks**	0.00	0.959	1.01	0.327	0.19	0.670	1.11	0.360	0.29	0.916	2.4	*** 0.042**	0.85	0.52
**Assigned Visits**	2.75	0.113	0.08	0.778	0.73	0.402	1.97	0.089	0.46	0.808	0.61	0.689	0.69	0.630
**Assigned Visits 1+ Lick**	0.76	0.395	0.12	0.731	0.36	0.554	0.81	0.549	0.3	0.911	2.12	0.069	1.06	0.387
**Percent Visits to Assigned Corner**	0.01	0.944	0.06	0.804	0.16	0.697	4.98	****** <0.0005**	0.18	0.971	1.48	0.203	0.24	0.946
**Percent Visits to Assigned Corner 1+ Lick**	0.01	0.904	0.00	0.945	0.16	0.694	3.58	****** 0.005**	0.14	0.983	2.72	0.024	0.77	0.575
**REVERSE PLACE PREFERENCE**														
	**ME Geno**		**ME Diet**		**Geno x Diet**		**ME Day**		**Geno x Day**		**Diet x Day**		**Geno x Diet x Day**	
	**F_(1, 19)_**	** *p* **	**F_(1, 19)_**	** *p* **	**F_(1, 19)_**	** *p* **	**F_(5, 95)_**	** *p* **	**F_(5, 95)_**	** *p* **	**F_(5, 95)_**	** *p* **	**F_(5, 95)_**	** *p* **
**Total Visits**	1.68	0.21	0.02	0.903	2.77	0.112	0.89	0.489	1.87	0.106	5.01	****** <0.0005**	0.99	0.425
**Total Licks**	0.16	0.691	2.36	0.141	0.1	0.755	9.38	****** <0.0005**	1.25	0.290	13.8	****** <0.0005**	1.2	0.315
**Assigned Visits**	2.94	0.103	1.38	0.255	4.01	0.06	1.76	0.128	0.68	0.64	8.43	****** <0.0005**	0.49	0.785
**Assigned visits 1+ Lick**	5.99	*** 0.024**	0.68	0.421	1.73	0.204	6.99	****** <0.0005**	1.58	0.173	12.86	****** <0.0005**	0.90	0.485
**Percent Visits to Assigned Corner**	0.01	0.914	1.17	0.293	0.41	0.528	1.41	0.227	1.73	0.135	1.7	0.143	1.72	0.137
**Percent Visits to Assigned Corner 1+ Lick**	0.19	0.667	0.19	0.664	0.40	0.535	1.86	0.109	3.50	**0.006**	2.12	0.069	0.90	0.482
**Percent Visits to Previous Corner (Day1)**	0.5	0.49	15.55	***** 0.001**	0.88	0.360								

**Table 2 nutrients-16-04167-t002:** Inflammatory cytokines in blood plasma. We evaluated endpoint blood plasma levels of 23 cytokines in a subset of animals (n = 3–5 per sex, diet, and genotype) using the Bio-Plex^®^ suspension array multiplexing system. Plasma from each animal was run in technical duplicates and averaged per animal. Here, we report the main effects (ME) of genotype (Geno) and diet from MANOVA of males and females separately. No significant interactions were observed. *p* values of ME below the threshold for significance (*p* < 0.05) are shown in bold: * *p* < 0.05.

**MALES**				
**Cytokine**	**ME Geno**		**ME Diet**	
	**F_(1, 14)_**	** *p* **	**F_(1, 14)_**	** *p* **
Eotaxin	0.00	0.998	2.95	0.108
Granulocyte colony-stimulating factor (G-CSF)	1.3	0.274	1.83	0.198
Granulocyte-macrophage colony-stimulating factor (GM-CSF)	1.64	0.221	1.60	0.226
Interferon γ (IFN-γ)	1.61	0.225	1.45	0.248
Interleukin (IL) 1α (IL-1α)	2.62	0.128	0.33	0.574
IL-1β	0.79	0.389	0.85	0.373
IL-2	1.05	0.324	0.72	0.410
IL-3	1.17	0.297	1.29	0.275
IL-4	1.30	0.274	0.98	0.339
IL-5	1.16	0.299	0.96	0.343
IL-6	1.87	0.193	0.92	0.355
IL-9	1.62	0.224	1.13	0.306
IL-10	2.62	0.128	0.33	0.574
IL-12p40	2.48	0.138	0.02	0.883
IL-12p70	1.50	0.242	0.58	0.461
IL-13	1.60	0.227	1.07	0.318
IL-17α	0.77	0.395	1.03	0.328
Chemokine (C-X-C motif) ligand 1 (CXCL-1)	2.16	0.164	0.41	0.531
Monocyte chemoattractant protein 1 (MCP-1)	1.77	0.204	0.48	0.501
Macrophage inflammatory protein 1α (MIP-1α)	2.07	0.172	1.33	0.268
Macrophage inflammatory protein 1β (MIP-1β)	1.61	0.226	0.86	0.369
Chemokine (C-C motif) ligand 5 (CCL-5)	1.40	0.256	0.00	0.984
Tumor necrosis factor α (TNF-α)	1.03	0.327	1.23	0.286
**FEMALES**				
**Cytokine**	**ME Geno**		**ME Diet**	
	**F_(1, 11)_**	** *p* **	**F_(1, 11)_**	** *p* **
Eotaxin	0.01	0.942	0.88	0.369
G-CSF	5.46	*** 0.039**	2.53	0.140
GM-CSF	3.52	0.087	4.99	*** 0.047**
IFN-γ	8.43	*** 0.014**	7.37	*** 0.020**
IL-1α	5.31	*** 0.042**	5.79	*** 0.035**
IL-1β	7.54	*** 0.019**	4.93	*** 0.048**
IL-2	5.96	*** 0.033**	3.51	0.088
IL-3	7.63	*** 0.018**	7.81	*** 0.017**
IL-4	3.92	0.073	2.21	0.165
IL-5	5.51	*** 0.039**	3.93	0.073
IL-6	5.36	*** 0.041**	4.90	*** 0.049**
IL-9	5.06	*** 0.046**	6.84	*** 0.024**
IL-10	8.02	*** 0.016**	4.13	0.067
IL-12p40	7.45	*** 0.020**	1.47	0.251
IL-12p70	7.88	*** 0.017**	6.38	*** 0.028**
IL-13	4.89	*** 0.049**	0.97	0.346
IL-17α	8.58	*** 0.014**	8.52	*** 0.014**
CXCL-1	4.40	0.060	3.77	0.078
MCP-1	3.95	0.072	3.46	0.090
MIP-1α	5.81	*** 0.035**	4.14	0.067
MIP-1β	4.90	*** 0.049**	4.49	0.058
CCL-5	4.56	0.056	3.19	0.102
TNF-α	7.35	*** 0.020**	6.35	*** 0.028**

## Data Availability

The original contributions presented in this study are included in the article/Appendix A. Further inquiries can be directed to the corresponding author.

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
