# Peer review of "Assessing the Benefit of Dietary Choline Supplementation Throughout Adulthood in the Ts65Dn Mouse Model of Down Syndrome"

_nutrients, 2024, doi:10.3390/nu16234167_

Round 1
Reviewer 1 Report
Comments and Suggestions for Authors
Dear Authors,
The study explored dietary Ch+ in adult Ts65Dn mice. I greatly appreciated the comprehensiveness of the methods and results presented in this study. Here are my comments:
MAIN TEXT and ABSTRACT:
- Avoid using "we" or "our" for a more formal tone.
ABSTRACT:
- Since references cannot be included in the abstract, rewrite the sentence referring to your previous study in a general manner, avoiding explicit mention of a specific study.
- The aim of the study is clear, however consider rephrasing it to ensure greater clarity, that is "Dietary Ch+ in adult Ts65Dn mice has not yet been explored...and so... the aim..." followed by your purpose.
- Include the most significant statistical outcomes in the results section of the abstract for a more complete summary.
INTRODUCTION:
- I suggest adding context on how various supplementations, such as choline, have been shown to improve cognitive functions and how others, like carnitine, can enhance motor performance (DOI: 10.3390/jfmk6040093) also in healthy subjects, so also in dfferent "categories" of disorders these substances should be explored. This underscores the ongoing interest in this area of research and highlights its relevance.
Author Response
Reviewer 1
ABSTRACT:
- Since references cannot be included in the abstract, rewrite the sentence referring to your previous study in a general manner, avoiding explicit mention of a specific study.
Thank you for this suggestion; we have updated the abstract accordingly.
- The aim of the study is clear, however consider rephrasing it to ensure greater clarity, that is "Dietary Ch+ in adult Ts65Dn mice has not yet been explored...and so... the aim..." followed by your purpose.
We appreciate your suggestion to improve clarity. We have slightly re-organized the text between the “Background/Objectives” and “Methods” sections to rephrase this and make it clearer.
- Include the most significant statistical outcomes in the results section of the abstract for a more complete summary.
This is a helpful suggestion to improve the impact of the abstract, thank you. The results discussed are the most significant, so we changed the wording to make this clear.
INTRODUCTION:
- I suggest adding context on how various supplementations, such as choline, have been shown to improve cognitive functions and how others, like carnitine, can enhance motor performance (DOI: 10.3390/jfmk6040093) also in healthy subjects, so also in dfferent "categories" of disorders these substances should be explored. This underscores the ongoing interest in this area of research and highlights its relevance.
Thank you for this suggestion. We have added a sentence including two new citations in the introduction (Lines 57-60) describing that supplements can improve health outcomes in healthy individuals, which further highlights the potential for these to also help individuals with impairments, for example in AD and DS.
Reviewer 2 Report
Comments and Suggestions for Authors
Dear Authors,
Your study's subject is current and covers an under-investigated area of neurological pathologies relevant to DS and AD.
Heren are my minor comments on your paper:
1. Please merge citations throughout the manuscript (for instance in line 50);
2. Some minor editorial corrections are necessary (typos, justified positions of the text – for instance line 314);
3. The results of the study are well-presented, I especially like the figures, which are very easy to read and informative;
4. I suggest using “older adults” instead of “the elderly” in line 682. The term ‘the elderly’ brings negative connotations to older people;
5. The study is well discussed and the results fully support the conclusions.
Indeed, it was a pleasure to read your manuscript.
Best regards,
The reviewer.
Author Response
Reviewer 2
Your study's subject is current and covers an under-investigated area of neurological pathologies relevant to DS and AD.
Heren are my minor comments on your paper:
- Please merge citations throughout the manuscript (for instance in line 50);
Thank you for correcting this. We had placed citations within the sentences to make clear which statement corresponds with which citation, but see now that the reference requirements state to place directly before the punctuation and to merge citations (e.g., from [80], [81] to [80,81]. We have updated all instances of this in the manuscript, excepting a few locations where we felt it critical for the references to show contrasting data (e.g., differential benefits of supplementary choline in Lines 83-91, or in males versus females in Line 98).
- Some minor editorial corrections are necessary (typos, justified positions of the text – for instance line 314);
Thank you for this – we have corrected the justification throughout the entire manuscript to make it consistent with the provided template. We have also added review from an independent scientist to make sure all typos and formatting inconsistencies have been found and addressed.
- The results of the study are well-presented, I especially like the figures, which are very easy to read and informative;
Thank you for this comment, it is much appreciated.
- I suggest using “older adults” instead of “the elderly” in line 682. The term ‘the elderly’ brings negative connotations to older people;
This is an important consideration, and we have updated our terminology accordingly.
- The study is well discussed and the results fully support the conclusions. Indeed, it was a pleasure to read your manuscript.
Thank you for this comment, we appreciate it and are glad that you enjoyed this work.